# Acute Effects of Augmented Reality Exergames versus Cycle Ergometer on Reaction Time, Visual Attention, and Verbal Fluency in Community Older Adults

**DOI:** 10.3390/ijerph192214667

**Published:** 2022-11-08

**Authors:** Soraia Ferreira, José Marmeleira, Jesus del Pozo-Cruz, Alexandre Bernardino, Nilton Leite, Mafalda Brandão, Armando Raimundo

**Affiliations:** 1Departamento de Desporto e Saúde, Escola de Saúde e Desenvolvimento Humano, Universidade de Évora, Largo dos Colegiais, 7000-727 Évora, Portugal; 2Comprehensive Health Research Centre (CHRC), Palácio do Vimioso, Gabinete 256, Largo Marquês de Marialva, Apart. 94, 7002-554 Évora, Portugal; 3Department of Physical Education and Sports, University of Seville, 41013 Sevilla, Spain; 4Epidemiology of Physical Activity and Fitness across Lifespan Research Group (EPAFit), University of Seville, 41013 Sevilla, Spain; 5Instituto de Sistemas e Robótica, Instituto Superior Técnico, 1049-001 Lisboa, Portugal

**Keywords:** executive functions, cognition, older adults, acute effects, augmented reality

## Abstract

Background: This study aims to investigate the acute effects of an augmented reality session and a cycle ergometer session compared to no exercise on the reaction times, cognitive flexibility, and verbal fluency of older adults. Methods: Each participant did a familiarization with cognitive tests and the following three sessions: cycle ergometer, no exercise (control group), and augmented reality exergame (Portable Exergame Platform for Elderly) sessions. The participants were randomized in a within-group design into one of six possible combinations. Each moment had a 30 min duration, and after the session, the participants performed a Trail Making Test, a verbal fluency test, and a Deary–Liewald reaction time task. The data were analyzed with a one-way ANOVA with a Bonferroni adjustment. Results: The analysis between the no exercise, cycle ergometer, and augmented reality sessions showed no significant differences in the cognitive measurements. Conclusions: One session of the cycle ergometer exercise or the augmented reality exergames does not acutely improve the reaction times, cognitive flexibility, or verbal fluency in the elderly.

## 1. Introduction

Over the past few years, researchers have been studying the acute effects of exercise and have found some interesting results. Acute aerobic exercise helps to increase cerebral blood flow in certain cortical areas [1,2], which could be influenced by the exercise intensity [2,3]. Brain oxygenation remains high up to 30 min after the end of an exercise; ideally, a cognitive task should be performed during this time [4]. Several studies have reported on the time required for one session to produce the acute effects of exercise on cognition. A study with women who had breast cancer showed that 30 min of moderate-intensity exercise improved their cognition [5]. Similar results were also observed in a study of elderly people with depression who exercised at moderate intensity for 30 min [6]. Twenty minutes of moderate- to vigorous-intensity physical activity has also been shown to affect cognition in adults [7]. A typical session of acute exercise includes an initial assessment of parameters (cognitive ability or physical ability), followed by 10 to 30 min of physical activity at an intensity greater than 40%, and a re-measurement of the initially measured parameters at the end [8]. In recent years, several research studies have been conducted to investigate the acute effects of exercise on cognitive abilities. Cognitive improvements have been noted in studies of young adults [9], patients with depression [6], breast cancer [5] and multiple sclerosis [10] patients, and older adults [11,12].

Few studies have observed the acute effects of exergame sessions in the elderly population. One study examined the acute effects of an Xbox Kinect session on the moods of the participants. The results of this study showed that an exergame session was associated with improved mood, just as the pleasure of playing while exercising was associated with improved mood [13]. In addition, two studies examined the effects of video games on cognition. The first study showed that reaction time and cognitive flexibility improved after a video game session in community-dwelling older adults [14]. The second study showed no significant differences in the cognitive abilities of older adults living in a nursing home [12]. Our study is premised on the need to understand the acute effects of augmented reality on the cognitive abilities of older adults. Therefore, we aim to investigate the acute effects of two different types of training—an augmented reality session and a bicycle ergometer session—on the reaction times, cognitive flexibility, and verbal fluency in older adults.

## 2. Materials and Methods

### 2.1. Participants

Twenty-eight volunteers living in the region of Évora (Portugal) were eligible for this study. To recruit participants for this study, flyers were placed in mailboxes. The subjects who were interested contacted the researchers. The inclusion criteria were as follows: living in a community, having no contraindications to moderate aerobic exercise, and being 60 years of age or older. The exclusion criteria were as follows: having a mild cognitive impairment or more. Cognitive impairment was considered according to the scores of the Portuguese version of the Mini-Mental State Examination (MMSE), with cut-offs of ≤27 points for persons with >11 years of school education, ≤22 for persons ranging from 1 to 11 years of school education, and ≤15 points for illiterate persons [15]. Table 1 shows the general characteristics of the participants.

All of the participants were informed about the study’s objectives and signed a consent form prior to their participation. The study was approved by the University of Évora Ethics Committee and conducted following the Declaration of Helsinki.

### 2.2. Procedure

In this study, all of the participants initially familiarized themselves with the cognitive tests. Subsequently, all of the subjects participated in three moments (cycle ergometer, augmented reality exergame, and non-exercise). The participants were randomized in a within-group design into one of six possible combinations (Figure 1), and the sessions for each moment were separated between 7 and 10 days. 

Each participant’s resting heart rate was recorded, and the maximal heart rate was determined using the following formula: HR max = 208.75 − 0.73 × age [16]. To define the intended exercise intensity (60% to 70%), the formula used was as follows: % HR = [(HR max − HR resting) × (% HR reserve)] + HR resting [17]. During the activity, each participant’s heart rate was monitored by a chest heart rate monitor.

In the cycle ergometer session, the participants exercised for 30 min while maintaining 50 rotations per minute. The resistance started at 50 W and was adjusted every two minutes until the intended level (HR was between 60% and 70%) was achieved.

The augmented reality session was conducted using exergames available on the Portable Exergame Platform for Elderly (PEPE) [18]. These exergames were designed to mobilize physical and mental resources simultaneously. Four (Exerpong, Grape stomping, Rabelos, and Toboggan games) of the five PEPE games were used during the 30-min training sessions, with HRs ranging from 60% to 70%. In the “Exerpong game” (Figure 2A), the participants had to perform lateral movements to move onto a platform (located at the bottom of the screen) and touch the ball moving through the scenario. The participants had to extend their arms when the ball hit the platform. In the “Rabelos game” (Figure 2C), the participants were on a boat travelling on the Douro River, and the goal was to move to the dock and pick up the barrels by latent movements. The participants had to move by rotating their arms and then pick up a barrel, making a squat exercise. Next, the participants had to try to stomp on as many grape bunches as possible in the “Grape Stomping game” (Figure 2D). Next, the participants had to step down from the three barrels projected on the floor and pull the grapes into the barrel by bending and stretching their arms. After placing the grapes in the barrel, the participants had to step on the grapes by alternately bending their knees. Finally, in the “Toboggan game” (Figure 2B), the players were placed inside wicker toboggans and instructed to move sideways while attempting to capture as many bananas as possible as they travelled down the street (which appeared during the descent).

The same definitions (difficulty level) were used for all of the participants in the exergames. The subjects who could not reach their target heart rate were given an external load, such as dumbbells, in their hands. In both of the training sessions, the Borg scale was also used to control the exercise intensity. In subjects taking beta-blockers, this indicator was also considered to adjust the external load. In the control group, the participants rested for 30 min and then performed the cognitive tests. The general procedure for the exercise sessions was similar to other studies [14,19] and was divided into the following phases: (1) measurement of blood pressure; (2) activation of HR monitor; (3) 30 min of activity; (4) immediate measurement of the Trail Making Test, verbal fluency, and reaction time. All of the sessions were conducted by an exercise physiologist. The application of the tests did not exceed 15 min.

### 2.3. Measures

The reaction time was measured by the Deary–Liewald reaction time task [20]. This task evaluated the simple reaction time (SRT) and the choice reaction time (CRT). For the SRT task, the spacebar key had to be pressed by the participants in response to a single stimulus. When the participant pressed the key, the cross inside the square disappeared and reappeared after a short time interval. Therefore, the participants had to press the spacebar key as quickly as possible after the stimulus. The task involved 2 practice trials and 20 test trials. The median of the SRT (ms) was computed for each participant. In the CRT task, four stimuli appeared, and there was a correct answer for each stimulus. For each answer, a different key was used. For example, the “z” key was used when a cross appeared in the square on the left, followed by the “x” key in the next square, the “comma” key in the next, and the “full-stop” key in the square on the right. The participants were required to press the key corresponding to the square that the cross appeared in as quickly as possible. Due to the participant’s unfamiliarity with the computer keyboard, the keys were highlighted in a different color. The task involved 4 practice trials and 20 test trials. The median of the CRT (ms) was used for the data analysis.

The Trail Making Test measured cognitive flexibility and processing speed [21]. This test consisted of two parts, A and B. In part A, the participants were asked to join the numbers from 1 to 25 in an ascending order as quickly as possible. In part B, the test consisted of numbers between 1 and 13 and letters between A and M. The participants were asked to join the numbers and letters alternately, connecting the 1-A-2-B-3-C and so on until the end. The score in both tests was the time that the participants took to put the numbers and letters together.

The phonemic and semantic verbal fluency test evaluated language production, non-motor processing speed, and executive functions. In the semantic verbal fluency test, the participants had to enumerate the most extensive number of animals in one minute. When the same animal was listed but with a different sex, it was not considered. In the phonemic verbal fluency task, the participants were asked to enumerate the maximum number of words starting with the letter P for 60 s. Words from the same family and proper names were not counted. The score in both tests was the number of correct words that the participants enumerated [22].

The cognitive variables were collected by an exercise physiologist who was familiar with all of the tests and was highly experienced in applying them.

### 2.4. Data Analysis

A descriptive analysis was used for all of the variables, observing the mean and standard deviation. The data normality was evaluated by the Shapiro–Wilk test, and according to these results, an ANOVA test was used. The statistically significant differences in the Trail Making Test, verbal fluency, and DLRT were analyzed using a one-way ANOVA test with Bonferroni post-hoc tests. Furthermore, a univariate ANCOVA was used to control the effects of the beta-blockers (a covariate) in the cognitive tests. For the cognitive tests, a composite score was calculated (z score). The analyses were conducted using PASW Statistical for Windows (Version 22.0; IBM SPSS Inc.). For all of the statistical tests, the significance was set at *p* < 0.05.

## 3. Results

Twenty-eight participants were recruited for this study. However, one of the participants was unable to complete the protocol and was omitted. Thus, 27 subjects (9 women and 18 men) participated in this study. Eighteen of the participants had hypertension, eight had diabetes, nine had cardiovascular disease, and seventeen had hypercholesterolemia. Twenty-five of the participants were taking medications for hypertension, diabetes, hypercholesterolemia, and cardiovascular disease, and nine of them were taking beta-blockers. All of the participants took part in both the control group (no exercise) and the augmented reality group. However, two of the participants could not perform the exercises on the cycle ergometer because they could not reach the pedals. As an alternative, they performed a similar exercise on a treadmill, meeting the same heart rate criteria. The comparison between the three groups in the cognitive tasks showed no significant differences (Table 2). There were no statistical differences (p’s > 0.05) between the CG and AR in the Trail Making Tests A and B, verbal fluency, and reaction time (simple and choice). When comparing the control group with the cycle ergometer, the results were the same as those presented previously. In the comparison between the augmented reality and the cycle ergometer, no significant differences were found in the Trail Making Tests A and B, verbal fluency, simple reaction time, and choice reaction time. The ANCOVA results also confirmed that there were no significant differences between the cognitive performances in the three moments when controlling for the use of beta-blockers.

The z score represented the composite of the cognitive tests and was calculated for each participant. No significant differences were found in the composite of the six cognitive tasks.

## 4. Discussion

This study aimed to investigate the acute effects of augmented reality and bicycle ergometer sessions on the reaction time, cognitive flexibility, and fluency of older adults. The results showed that there were no significant differences in the cognitive tasks after 30 min of rest or physical activity with the cycle ergometer or augmented reality.

One study observed the acute effects of aerobic exercise, video games, and the combination of aerobic exercise and video games on cognitive flexibility and choice reaction time [14]. The older adults showed improvements in choice reaction time and cognitive flexibility at all times of the video game session, with cognitive flexibility displaying superior results [14]. Another study with older adults examined the acute effects of a cycle ergometer compared with a session of no exercise [11]. The authors examined the changes in reaction times after a 20-min session; in contrast to our study, the reaction time was shorter after the cycle ergometer session [11]. In this study, the participants’ work rates corresponded to 30% and 50% of the maximal oxygen uptake (VO_2_ max).

To the best of our knowledge, there are a few studies investigating the acute effects of augmented reality exergames in older people. However, we did find a few studies with younger participants. For example, one study conducted with young adults concluded that aerobic exercise and aerobic training combined with video games had positive effects on decision reaction times. On the other hand, the control group, which was at rest, showed no improvements in their choice reaction time [23]. Another study investigated the acute effects of watching a video and exercising with a cycle ergometer [24]. This investigation compared the acute effects of a 20-min exercise on a cycle ergometer and watching a 45-min video about the benefits of physical activity in patients with chronic obstructive pulmonary disease. Before and after the session, four cognitive tests—the Trail Making Test (cognitive flexibility and processing speed), digit symbol (psychomotor performance), verbal fluency (verbal processing), digit span (attention and short-term memory), and finger tapping (motor speed)—were completed. After the exercise session, patients with chronic obstructive pulmonary disease showed improvements in their verbal expressiveness, but watching videos did not improve their cognitive abilities [24].

Several underlying mechanisms have been associated with cognitive performance during acute aerobic exercise, including changes in catecholamines, serotonin, and the brain-derived neurotrophic factor (BDNF) [25,26]. In specific, it is important to note that BDNF promotes neuronal growth and synaptic plasticity [27]. One study examined the relationship between plasma catecholamine and cognitive performance during acute aerobic exercise. This type of training was found to have no effect on the executive control processes [11]. On the other hand, in another study, aerobic training was found to affect executive control functions at the neuroelectric level [28]. Another evidence is the decreased reaction times and the beneficial effects on long-term memory with moderate and vigorous aerobic training [29,30].

In acute dual-task exercises, the eventual brain and cognition benefits could also be due to perceptive–cognitive mechanisms, including divided attention and additional focus. It was shown that acute training with exergames improved cognitive and perceptual functions and executive function controls [14]. Another study showed that activities involving multiple cognitive processes can increase the transfer of cognitive performance [31]. The fact that a cognitive task is associated with physical activity may induce a higher level of arousal that promotes learning [32]. Other aspects associated with exergames that may influence cognitive functions are the challenges and reward systems of the game and the virtual environments [12]. However, the mechanisms underlying cognitive functions and exergames are not well defined yet.

In contrast to previous results and in line with our findings, similar studies did not show improvements in reaction time, short-term memory, executive functions, semantic memory, and working memory after a training session with exergames [12,23]. In the study of Monteiro-Junior [12], conducted with institutionalized older adults, there were no differences between the control group and the group using the Nintendo Wii in the tests of verbal fluency, forward digit span, and backward digit span. However, in Monteiro-Junior’s study, the Nintendo Wii group’s semantic memory and executive functions improved over time.

There are some reasons that could have influenced our results. Some of the participants felt so tired that they could not reach their target heart rate. Therefore, the effort perception scale was used to assess how fatigued one felt on a personal level. The participants who took beta-blocking medication and were very tired, even when their heart rate was still not within the defined values, decreased their exercise intensity. Another aspect that may have influenced our results was the lack of cognitive testing before and after the sessions. In our study, there were four different time points (familiarization, augmented reality, cycle ergometer, and a control group), which could cause a learning effect that was not intended because each participant completed the tests three times. 

We found some limitations, namely the fact that the sample was small and that there was no control for the history of diseases (e.g., cardiovascular problems) that could affect cognitive performance. To the best of our knowledge, this is the first study to examine the immediate effects of a single augmented reality session on the cognitive abilities of older people. Future studies are important to understand the acute effects of augmented reality on the cognitive abilities of the elderly.

## 5. Conclusions

The single augmented reality session showed no significant improvements compared to the session with the cycle ergometer and without exercise in verbal fluency, reaction time, and cognitive flexibility. For future studies, the sample should be more homogeneous in terms of medication use, as beta-blocker use does not allow for accurate heart rate control.

## Figures and Tables

**Figure 1 ijerph-19-14667-f001:**
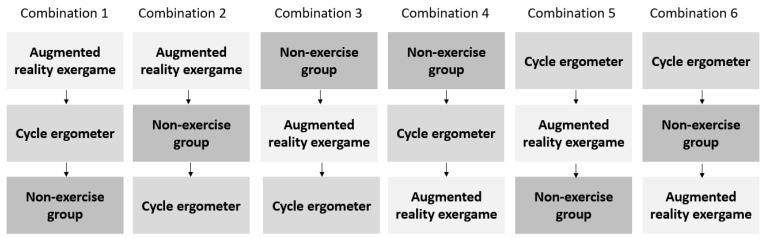
Different combinations of exercise implementation.

**Figure 2 ijerph-19-14667-f002:**
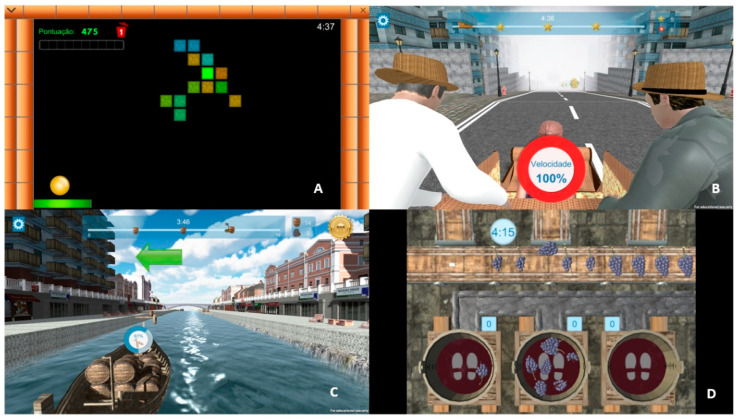
Different games on the Portable Exergame Platform for Elderly. (**A**) Exerpong; (**B**) Toboggan games; (**C**) Rabelos; (**D**) Grape stomping.

**Table 1 ijerph-19-14667-t001:** Descriptive characteristics of the participants.

	Participants Mean (SD)
Age (years)	72.0 (5.2)
Weight (kg)	75.8 (11.5)
Height (cm)	161.9 (8.7)
BMI (kg/m^2^)	28.9 (3.6)
MMSE (points)	27.9 (1.4)
Education (years)	8.1 (5.3)
Heart Rate (bpm)	65.0 (11.7)
Blood Pressure Systolic (mm Hg)	137.9 (15.2)
Blood Pressure Diastolic (mm Hg)	77.4 (10.2)

Note: BMI, Body Mass Index; MMSE, Mini-Mental State Examination.

**Table 2 ijerph-19-14667-t002:** Comparison between the cognitive tasks in the three different moments of evaluation.

	CG	AR	CE	*p **
Trail Making test A (score)	39.17 (17.9)	42.05 (17.1)	46.36 (29.5)	0.226
Trail Making test B (score)	145.39 (81.3)	126.31 (78.9)	157.08 (119.2)	0.331
Semantic fluency task (n)	16.85 (5.8)	16.96 (4.8)	17.78 (5.9)	0.882
Phonemic fluency task (n)	9.59 (4.7)	9.81 (5.2)	10.56 (5.5)	0.919
SRT (ms)	354.31 (45.8)	362.06 (47.5)	343.96 (44.9)	0.540
CRT (ms)	725.80 (174.6)	722.13 (156.4)	696.19 (159.5)	0.871
Cognitive composite (z-score)	−0.23 (0.1)	−0.03 (0.1)	−0.14 (0.1)	0.999

Note. AR, Augmented Reality; CG, Control Group; CE, Cycle ergometer; SRT, simple reaction time; CRT, choice reaction time; * *p* < 0.05 for comparison between the groups, according to a one-way ANOVA test with Bonferroni post-hoc tests.

## Data Availability

All data are presented in the manuscript.

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
