# Peer review of "Acute Effects of Augmented Reality Exergames versus Cycle Ergometer on Reaction Time, Visual Attention, and Verbal Fluency in Community Older Adults"

_ijerph, 2022, doi:10.3390/ijerph192214667_

Round 1
Reviewer 1 Report
Dear authors.
I find this article on the Acute effects of augmented reality exergames versus cycle ergometer on reaction time, visual attention, and verbal fluency in community older adults very interesting. The article is well evidenced and written.
But I find an important problem, the sample size is too small. The authors do not perform a sample size calculation. I recommend that you do a sample size calculation and be more cautious with your conclusions.
Who has carried out the intervention sessions? Who has passed the different cognitive tests?
Reviewer 2 Report
The purpose of this study was to evaluate the acute effects of an augmented reality session and a cycle ergometer session, in comparison to no exercise, on reaction time, cognitive flexibility, and verbal fluency in elderly people. The analysis not shown significant differences in cognitive measurements between: exercise, cycle ergometer and augmented reality.
Comments and Suggestions for Authors
1) Materials and Methods: The author report that: “The exclusion criteria were having a mild cognitive impairment or more “. But: history of hypertension been taken? cardiovascular problems? Was a baseline ECG performed prior to enrollment?
Did the subjects take: drugs/medication/supplementation/antioxidant/vitamins? All this is missing.
Table 1. Physiological data of SBP/DBP mmHg, SaO2, HR,?? Insert.
Also show the separate data of males and females.
- Insert short description test: Trail Making Test, Digit Symbol, Verbal Fluency, Digit Span, and Finger Tapping.
In Procedure: to help the reader insert images representing the used scenarios.
2) Results:
3.1….” Twenty-eight participants were recruited for this study. However, one of the participants could not finish the protocol and was omitted. Thus, twenty-seven subjects (9 women and 18 men) participated in this study. All participants participated in the control (without exercise) and the augmented reality sessions. However, two participants were unable to perform exercise on the cycle ergometer, as they did not reach the pedals. In the alternative, they performed a similar exercise using a treadmill, with the same heart rate criteria. …”.
For a greater scientific rigor, even the 2 subjects who did not reach the pedals were eliminated. So, the question is if you eliminate these 2 subjects, are the final results always the same?
There are differences in test between males and females?
3) Discussion:
The authors report that: “In our study, some participants felt so tired that they could not reach their target heart rate. Therefore, the effort perception scale was used to assess how fatigued one felt on a personal level. Participants who took beta-blocking medication and were very tired, even when their heart rate was still not within the defined values, decreased the exercise intensity”.
Beta-blocking medication! This is an important factor that could have an effect on the final result, and could also be taken into account in statistics; as well as the absence of cognitive tests before and after the session.
Please argue more, the discussion is poor. Please explain better the results obtained with the cognitive tests in relation to different combinations of exercise.
Finally, perhaps it would be useful to add limitations to the study.
Round 2
Reviewer 2 Report
All the questions have been answered
best regards